# Effect of non-crop vegetation types on conservation biological control of pests in olive groves

Daniel Paredes[1], Luis Cayuela[2], Geoff M. Gurr[3] and Mercedes Campos[1]

[1] Grupo de Protección Vegetal, Departamento de Protección Ambiental, Estación Experimental de Zaidín, CSIC, Granada, Spain
[2] Área de Biodiversidad y Conservación, Departamento de Biología y Geología, Universidad Rey Juan Carlos I, Móstoles, Spain
[3] EH Graham Centre for Agricultural Innovation (Primary Industries, NSW and Charles Sturt University), Orange, NSW, Australia

## ABSTRACT

Conservation biological control (CBC) is an environmentally sound potential alternative to the use of chemical insecticides. It involves modifications of the environment to promote natural enemy activity on pests. Despite many CBC studies increasing abundance of natural enemies, there are far fewer demonstrations of reduced pest density and very little work has been conducted in olive crops. In this study we investigated the effects of four forms of non-crop vegetation on the abundance of two important pests: the olive psyllid (*Euphyllura olivina*) and the olive moth (*Prays oleae*). Areas of herbaceous vegetation and areas of woody vegetation near olive crops, and smaller patches of woody vegetation within olive groves, decreased pest abundance in the crop. Inter-row ground covers that are known to increase the abundance of some predators and parasitoids had no effect on the pests, possibly as a result of lack of synchrony between pests and natural enemies, lack of specificity or intra-guild predation. This study identifies examples of the right types of diversity for use in conservation biological control in olive production systems.

## INTRODUCTION

The use of synthetic pesticides for pest control in conventional agriculture is coming under closer scrutiny due to rising concerns about environmental and health problems (*Meehan et al., 2011*) including the recent ruling by the European Union to suspend the use of neonicotinoid insecticides (*Stokstad, 2013*). Biological control has evolved in recent decades as a response to such concerns and one approach, 'conservation biological control' includes practices such as the modification of the environment to boost the impact of natural enemies of pests (*Eilenberg, Hajek & Lomer, 2001*). The modification of the environment can involve features on or close to farms (*Boller, Häni & Poehling, 2004*), such as hedges, woodland patches, grasslands, wildflower strips, ruderal areas,

Corresponding author
Daniel Paredes,
daniel.paredes@eez.csic.es

conservation headlands and even stone heaps. The extent of spill over of natural enemies between adjacent vegetation types can be large (*Bowie et al., 1999*). These various forms of modifications increase the diversity of vegetation and habitat structure, which in turn increases the availability of natural enemy resources such as nectar, pollen, alternative prey, and shelter (*Altieri & Letourneau, 1982*; *Landis, Wratten & Gurr, 2000*).

The increase in natural enemy biodiversity that is often reported in studies with supplementary non-crop vegetation (*Bianchi, Booij & Tscharntke, 2006*; *Thies & Tscharntke, 1999*) can potentially increase ecosystem function in the form of top-down forces from the third to the second trophic level (*Landis, Wratten & Gurr, 2000*). Yet, the extent to which top-down forces are effectively activated by non-crop vegetation is variable and often not reported in papers that achieve effects purely at the third trophic level. Some studies have reported a decrease in herbivore density in the presence of ground cover vegetation (*Aguilar-Fenollosa et al., 2011*; *Altieri & Schmidt, 1986*) or patches of natural vegetation in and around farms (*Thies & Tscharntke, 1999*; *Thomson & Hoffmann, 2009*), whilst others have found no effect (*Bone et al., 2009*; *Danne et al., 2010*). Lack of effect can be the result of intra-guild predation (*Polis, Myers & Holt, 1989*; *Straub, Finke & Snyder, 2008*), disruption of biological control by alternative prey presence (*Koss & Snyder, 2005*) or asynchrony between herbivores and their natural enemies (*Fagan et al., 2002*; *Perdikis, Fantinou & Lykouressis, 2011*). Additionally, there is the risk that pest species might be attracted by, and make use of non-crop vegetation, thus increasing – rather than decreasing – their abundance (*Baggen & Gurr, 1998*; *Gurr, van Emden & Wratten, 1998*; *Wratten & van Emder, 1995*). The foregoing factors underline the importance of identifying the right kinds of diversity for use in any conservation biological control program.

This study aimed to compare the relative benefits of four forms of vegetation diversity for use in olive grove conservation biological control (inter-row ground covers, areas of herbaceous vegetation and areas of woody vegetation near olive crops, and smaller patches of woody vegetation within olive groves) on the abundance of olive moth (*Prays oleae* (Bernard)), and olive psyllid (*Euphyllura olivina* (Costa)).

There is a strong need for research on olive pests because world economic losses are estimated at €800 million per year, with an additional cost of €100 million year in agrochemical products (*IOBC, 2005*). These figures do not include the potential environmental and health costs of insecticide use. Research on the potential for conservation biological control approaches to lessen reliance on insecticide use on olive pests is scarce (*Herz et al., 2005*). A recent study in this system revealed that ground cover increased the abundance of spiders, Hymenopteran parasitoids and one species of predatory Heteroptera (*Deraeocris punctum*) (*Paredes, Cayuela & Campos, 2013*). If alternative forms of vegetation diversity can reduce pest abundance, the adoption of these by olive growers could alleviate the associated cost of conventional pest control, whilst potentially also improving other ecosystems services such as soil fertility, prevention of soil erosion (*Cullen et al., 2008*; *Hartwig & Ammon, 2002*) and pollination (*Tscheulin et al., 2011*).

## MATERIAL AND METHODS

### Study species

In this study we focused on the nymphs of *E. olivina* and the adults of the flower generation of *P. oleae*. The nymphs of *E. olivina* appear from mid-April to the end of May and, during this period, are susceptible to the attack by natural enemies. The flower generation of *P. oleae* is the most abundant of the three generations, and lays the eggs of the fruit generation. There is a strong correlation between the adults of the flower generation and the degree of olive fruit infestation (*Ramos et al., 1989*), and therefore it is assumed that controlling the population of the flower generation of *P. oleae* might ultimately reduce fruit infestation. Larvae of the flower generation of *P. oleae* appear at the same time as that of *E. olivina* nymphs. Other pests such as *Saissetia oleae* (Olivier) or *Bactrocera oleae* (Gmelin), that are potentially serious insects for the olive culture, were not included in this study because their attack was very low for the first in the study area. *B. oleae* used to appear from September to November, a period out of the target in this study. Several potential natural enemies of *E. olivina* and *P. oleae* were found in the experimental orchard. The most abundant were the spider families Thomisidae, Philodromidae, Araneidae, Salticidae, Linyphiidae and Oxyopidae; the parasitoid families Scelionidae, Encyrtidae, Elasmidae and Braconidae; the ant genus Tapinoma sp., Camponotus spp. and Plagiolepis sp.; the predatory Heteropteran species *Brachynotocoris ferreri* n. sp. Baena (in litteris), *Pseudoloxops coccineus* (Meyer Dur), *Deraeocoris punctum* (Rambur) and *Anthocoris nemoralis* (Fabricius); the green lacewing species *Chrysoperla carnea* (Stephens); and the order Raphidioptera.

### Vegetation treatments

The study was conducted in an experimental olive grove (235 ha) located in southern Spain, near the city of Granada (37°17′ N and 3°46′ W). Full information about the study area can be found in *Paredes, Cayuela & Campos (2013)*. Four different forms of non-crop vegetation were investigated: inter-row ground covers, areas of herbaceous vegetation and areas of woody vegetation near olive crops, and smaller patches of woody vegetation within olive groves (Fig. 1). Effects of the inter-row ground covers treatment were investigated by establishing 12 square-shaped plots each containing a grid of $7 \times 7$ olive trees and an area of 4900 m$^2$. Plots were separated by 150 m from each other. Half of the plots were treated with glyphosate and oxyflourfen in early spring 2010 and 2011 to remove weeds and provide a bare soil treatment. In the remaining plots, spontaneous herbaceous vegetation was allowed to grow in a 2.5 m wide strip between tree lines (Fig. 1A). Climatic conditions were different in both years. Average annual precipitation was higher in 2010 (565.12 mm) than in 2011 (368.82 mm). In 2010 mean average temperature was lower (22.1°C) than in 2011 (24.5°C). Average maximum monthly temperature from April to June was 24.7°C in 2010 and 27.8°C in 2011. Ground cover was composed of herbaceous plants dominated by *Medicago minima* L., *Anacyclus clavatus* Desf., *Hordeum leporinum* L., *Lolium rigidum* Gaudich., and *Bromus madritensis* L. The area beside the groundcover strips were treated with the same herbicides used for the bare soil treatment. No insecticides were used in the

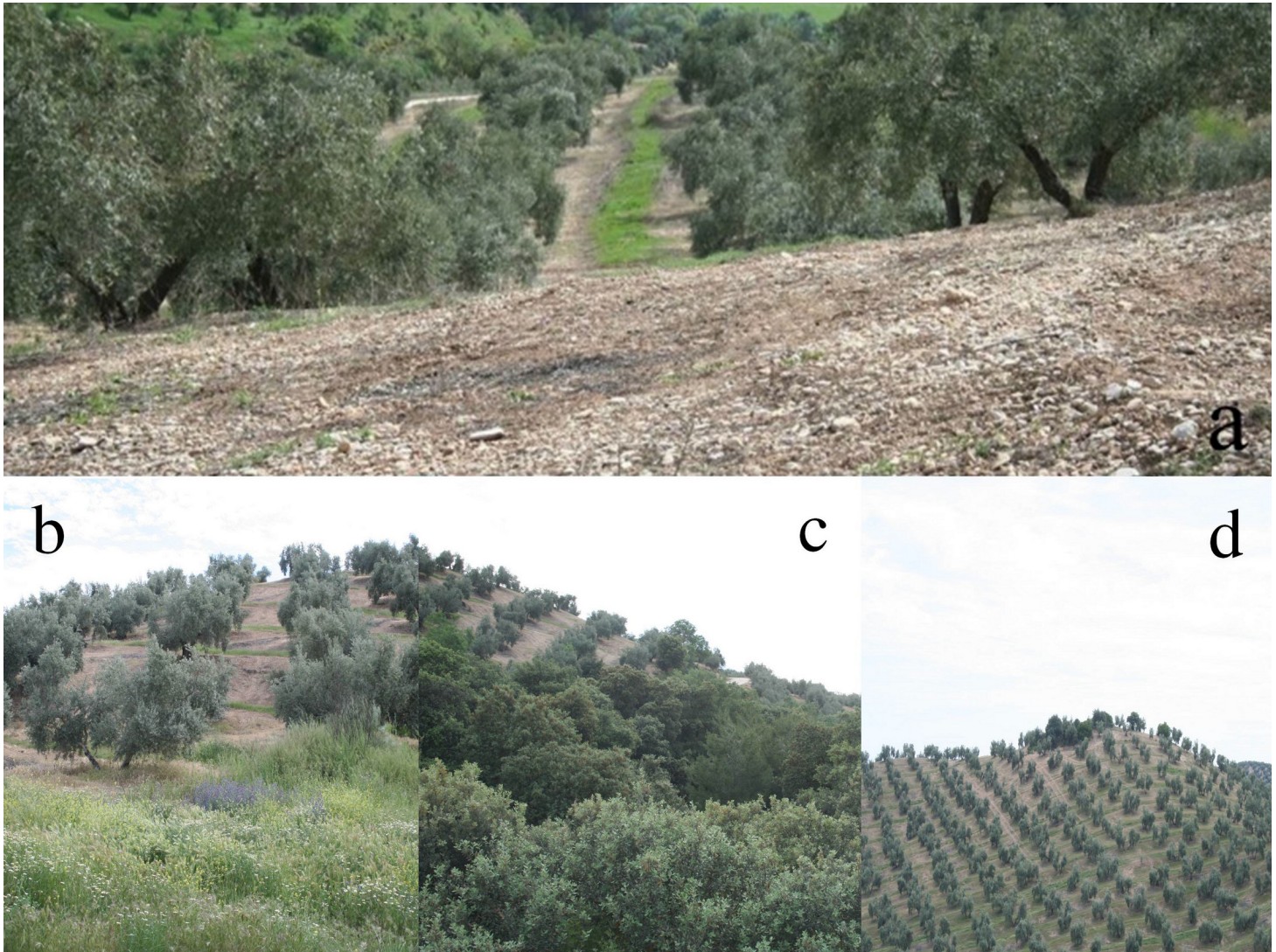

**Figure 1 Vegetation treatments.** (A) inter row-strips with cover plants (ground cover) and bare soil; (B) herbaceous vegetation adjacent to olive groves; (C) large woody vegetation adjacent to olive groves; (D) small woody patches within olive groves at the top of a hill.

grove for two years before the beginning of the experiment onwards. Other treatments were naturally occurring vegetation patches.

Herbaceous vegetation was adjacent to some of the olive areas and dominated by *Anchusa* sp., *Anacyclus clavatus* Desf. and *Echium plantagineum* L (Fig. 1B). Likewise, large woody patches were present beside some areas of olives as relatively extensive areas, partly in ravines, and comprised predominantly of *Phyllirea angustifolia* L. and *Quercus rotundifolia* Lam (Fig. 1C). Finally, small woody patches were comprised of shrubby vegetation patches occupying no more than a few square meters and located within the olive area, usually at hilltops, in areas of difficult access to machinery (Fig. 1D). This community consisted mainly of *Genista hirsuta* M. Vahl, *Cistus albidus* L., *Cistus clusii* Dunal, and *Rosmarinus officinalis* L.

Arthropod samples were collected every ten days, weather permitting, from late March to early July in 2010 and from early April to early July in 2011 giving 12 sampling dates in 2010 and nine dates in 2011. Sampling employed a modified vacuum device, CDC Backpack Aspirator G855 (John W. Hock Company, Gainesville, Florida, USA), which was used for two minutes per tree. On each date 16 olive trees (a grid of 4 × 4) were randomly selected for sampling. Arthropods were stored on ice for transportation to the laboratory, where they were identified and counted.

To incorporate the effect of each type of vegetation in models of *E. olivina* and *P. oleae* abundance (see below), we digitized all patches of non-crop vegetation using aerial photographs, and a 1 × 1 m resolution grid was superimposed upon the resulting vector maps. A bivariate Gaussian kernel density function (*Diggle, 1985*) was computed from the 1 × 1 m grid and a value was calculated for each plot in order to obtain a distance-weighted measure of the influence of each vegetation type on that plot. As we hypothesize that pest densities are going to be mediated by trophic interactions with their potential natural enemies, a standard deviation of 120 m was chosen for the Gaussian kernel density function, as *Miliczky & Horton (2005)* reported this as the maximum distance of dispersal of some groups of natural enemies in orchards. This measure also allowed incorporating the influence of vegetation at distances further than 120 m by assigning progressively lower weights, by means of the Gaussian density function, as we moved away from the plot. These analyses were performed with the R package 'spatstat' (*Baddeley & Turner, 2005*).

## Data modelling

A Gaussian function was used to describe the predicted abundance of the pests in response to time, since these insects typically show an increase in abundance during spring, reaching a maximum and declining afterwards. The Gaussian curve is a characteristic symmetric "bell curve" shape defined by three parameters: *a*, *b* and *c*. Parameter *a* is the height of the curve's peak; in our case, the maximum abundance that a certain pest could reach. The point in time when the highest abundance is reached is represented by parameter *b*, which is days since the first of January, and represents the mean of the Gaussian curve. Parameter *c* represents the standard deviation and controls the width of the "bell". The simplest model is represented by the equation:

$$\text{Abundance} = a \cdot \exp(-((X - b)^2 / 2 \cdot c^2))$$

where *X* is time, measured in days.

We added to the function a set of new parameters (*d.herb*, *d.lwp*, *d.swp*) that reflect the effect of herbaceous, large and small woody vegetation density on pest abundance. These account for the effect of each vegetation type on the estimated curve's peak (parameter *a*). A positive or negative effect means an increase or a decrease of pest abundance, respectively. The basic Gaussian function was modified to account for differences in the estimated parameters across plot treatments (ground cover and bare soil; parameter *a* and *d*) and years (2010, 2011; parameters *a*, *b*, *c* and *d*). We used likelihood methods and model selection as an alternative to traditional hypothesis testing (*Johnson & Omland, 2004*;

*Canham & Uriarte, 2006*), for data analysis. Following the principles of likelihood estimation, we estimated model parameters that maximized the likelihood of observing the abundance measured in the field, given a suite of alternative models. Overall, we tested 60 models, 20 more than in the former study due to we allow the parameter *d* to vary between years (for further details see *Paredes, Cayuela & Campos, 2013*). The most complex model, taking into account the effects of ground cover, natural vegetation, the interaction between these two, and inter-annual variability, can be expressed as:

$$\text{Abundance} = (a_{ij} + \gamma_{ij}) \cdot \exp(-((X - b_i)^2 / 2 \cdot c_i^2))$$

where $\gamma_{ij}$ represents the effect of natural vegetation patches on the maximum abundance of pests in each treatment (*j*) and year (*i*), which results from multiplying parameters $d.herb_{ij}$, $d.lwp_{ij}$ and $d.swp_{ij}$, estimated separately for bare soil and ground cover each year, by the observed values of herbaceous (*Xherb*), large (*Xlwp*) and small woody patches (*Xswp*) density, respectively, according to the following expression:

$$\gamma_{ij} = (d.herb_{ij} \cdot Xherb + d.lwp_{ij} \cdot Xlwp + d.swp_{ij} \cdot Xswp).$$

We used simulated annealing, a global optimization procedure, to determine the most likely parameters (i.e., the parameters that maximize the log-likelihood) given our observed data (*Goffe, Ferrier & Rogers, 1994*). We used a Poisson error structure for all response variables. Alternative models were compared using the Akaike Information Criterion ($AIC_c$) corrected for small sample size (*Burnham & Anderson, 2002*). Models with a difference in $AIC_c > 2$ indicate that the worst model has virtually no support and can be omitted from further consideration. We used asymptotic two-unit support intervals to assess the strength of evidence for individual maximum likelihood parameter estimates (*Edwards, 1992*). The $R^2$ of the model fit ($1 - SSE/SST$, sum of squares error (SSE) sum of squares total (SST)) of observed versus predicted was used as a measure of goodness-of-fit. All analyses were performed using the 'likelihood' package (*Murphy, 2012*) written for the R environment (*R Development Core Team, 2012*). R codes can be found in Appendix S1.

Based on best model predictions, we calculated the proportional change in pest abundance under the influence of a single vegetation type ($PCA_{VT}$) as follows:

$$PCA_{VT} = \Delta A_{VT} / \text{Maximum abundance}$$

where $\Delta A_{VT}$ is the difference between the predicted abundance in the presence of ground cover or maximum vegetation density for a particular vegetation type and the predicted abundance in the absence of any ecological infrastructure. $PCA_{VT}$ ranks from 1 to $-1$, reflecting the proportional increase (positive values) or decrease (negative values) of pest abundance under the influence of a particular vegetation type.

## RESULTS

A total of 7,530 insects were trapped during the study. Of these 4,940 were adults of *P. oleae* and 2,590 were nymphs of *E. olivina*. For both of these major pests, the best model

**Table 1 Comparison of alternative models (using AIC$_c$) for the pests tested in the study.** The best model (lowest AIC$_c$) is indicated in boldface type. The number of parameters and $R^2$ refer to the best model. For brevity, we have presented in Table 1 only the most parsimonious (i.e., lowest AIC$_c$) of all possible models that include inter-annual variability with one, two or three parameters varying between years.

| | | | AIC$_c$ | |
|---|---|---|---|---|
| | | | *E. olivina* | *P. oleae* |
| Basic model | No vegetation effect | General response | 2497.99 | 2270.00 |
| | | Interannual variability | 1829.50 | 879.19 |
| | Natural vegetation effect | General response | 2310.68 | 2248.13 |
| | | Interannual variability | **1693.24** | **862.91** |
| Ground cover model | No vegetation effect | General response | 2437.80 | 2266.97 |
| | | Interannual variability | 2050.68 | 1639.93 |
| | Natural vegetation effect | General response | 2239.37 | 2247.25 |
| | | Interannual variability | 2237.08 | 2121.30 |
| | Natural vegetation effect × cover crop | General response | 2263.38 | 2255.42 |
| | | Interannual variability | 1852.42 | 1621.31 |
| | | $R^2$ Best Model | 0.60 | 0.95 |
| | | Number of Parameters Best Model | 12 | 12 |

included the effect of vegetation and inter-annual variability in the parameters *a*, *b*, *c* and *d* (*E. olivina*: $R^2 = 0.60$; *P. oleae*: $R^2 = 0.95$; Table 1). Ground cover did not have any effect on the abundance of pest species and consequently we did not calculate the proportional change in abundance for ground cover versus bare soil. Both species displayed differences in abundance between years. *Euphyllura olivina* displayed a two-fold lower abundance in 2010 (45 individuals/plot) than in 2011 (94 individuals/plot) (Table 2). For *P. oleae* the opposite temporal trend was apparent with 213 individuals/plot in 2010 but only 90 individuals/plot the following year (Table 2).

The effect of vegetation varied among pests and years (Table 2). Proportional change in abundance (PCA$_{VT}$) allowed quantifying the relative effect of each vegetation type on pest abundance, whether positive or negative (Fig. 2). Herbaceous vegetation had a negative effect on the abundance of both pests in both years (Fig. 2). For *P. oleae* this effect was associated to a reduction in abundance of ca. 20% in 2010, but the reduction was much smaller in 2011 (Fig. 2). For *E. olivina* there was a reduction in abundance of ca. 20% in both years (Fig. 2). Small and large woody vegetation was associated to opposing responses for the two pests. For *E. olivina,* the influence of small woody vegetation was associated to a reduction the population by 16% and 59% in 2010 and 2011, respectively (Fig. 2). For *P. oleae* there was a negligible effect in 2010 and an estimated increase of ca. 12% in 2011 (Fig. 2). Large woody vegetation was associated to a reduction of *P. oleae* by 19% and 11% in 2010 and 2011, respectively, but had no effect in 2010 and a positive effect in 2011 with *E. olivina* increasing in abundance by 13% (Fig. 2).

## DISCUSSION

Non-crop vegetation can have the effect of suppressing pest populations by increasing the abundance of different groups of natural enemies (*Thies & Tscharntke, 1999*; *Landis, Wratten & Gurr, 2000*; *Boller, Häni & Poehling, 2004*; *Bianchi, Booij & Tscharntke, 2006*)

**Table 2 Parameter estimates and two-unit support intervals (in brackets) for the most parsimonious models of abundance of the pests tested in this study.** Parameter *a* is the maximum abundance of the pest in each year (2010, 2011); *b* and *c* represent the mean and standard deviation of the Gaussian curve in different years. The parameters *d.herb*, *d.lwp* and *d.swp* represent the effect of herbaceous, large and small woody natural vegetation on the maximum abundance of the pests in each year.

|  |  | *E. olivina* | *P. oleae* |
|---|---|---|---|
| a | 2010 | 45 [48; 53] | 213 [209; 220] |
|  | 2011 | 99 [96; 102] | 90 [86; 95] |
| b | 2010 | 129 | 169 |
|  | 2011 | 120 | 162 |
| c | 2010 | 0.102 [0.098; 0.104] | 0.152 [0.151; 0.154] |
|  | 2011 | 0.090 [0.087; 0.093] | 0.178 [0.172; 0.184] |
| d.herb | 2010 | −1474.32 [−1956.04; −871.91] | −6097.19 [−7429.27; −4388.74] |
|  | 2011 | −2276.73 [−2972.59; −1574.81] | −707.05 [−1869.73; −602.28] |
| d.lwp | 2010 | −1034.20 [−2683.31; −909.83] | −14650.07 [−18665.02; −9521.84] |
|  | 2011 | 5164.31 [2588.72; 7543.65] | −3797.50 [−6992.69; 348.83] |
| d.swp | 2010 | −11190.67 [−5847.34; 15541.83] | −5691.26 [−18370.29; 7631.83] |
|  | 2011 | −86071.63 [−79770.35; 92788.34] | 18960.74 [8730.12; 29823.10] |

provided that the right types of vegetation are identified and promoted for use. In this study herbaceous and woody vegetation within or adjacent to olive groves offered scope to reduce pest abundance through an indirect effect rule by the presence of natural enemies. Specifically, herbaceous vegetation was associated with a consistent reduction of the abundance of both pests though this effect was slight for *P. oleae* in 2011 when this pest was relatively rare. Small woody areas within the olive crop were associated with a reduction of *E. olivina* in both years but especially in 2011 when this pest was relatively common, giving a reduction of nearly 60%. For *P. oleae*, small woody areas gave a significant reduction in 2010 when this pest was relatively common but a modest increase in the following year when its density was relatively low. In contrast, the influence of large woody vegetation patches led to decreased pest abundance except for *E. olivina* in 2011 when this pest was relatively common. The consistent effect found in this study (large woody vegetation on the abundance *P. oleae* and small woody vegetation on the abundance *of E. olivina*) could be due to large uncultivated areas being a better source of natural enemies than small areas (*Tscharntke et al., 2008*) but small areas distributed within the crop may facilitate natural enemies to move quickly into the crop (*Bianchi, Booij & Tscharntke, 2006*). As *P. oleae*

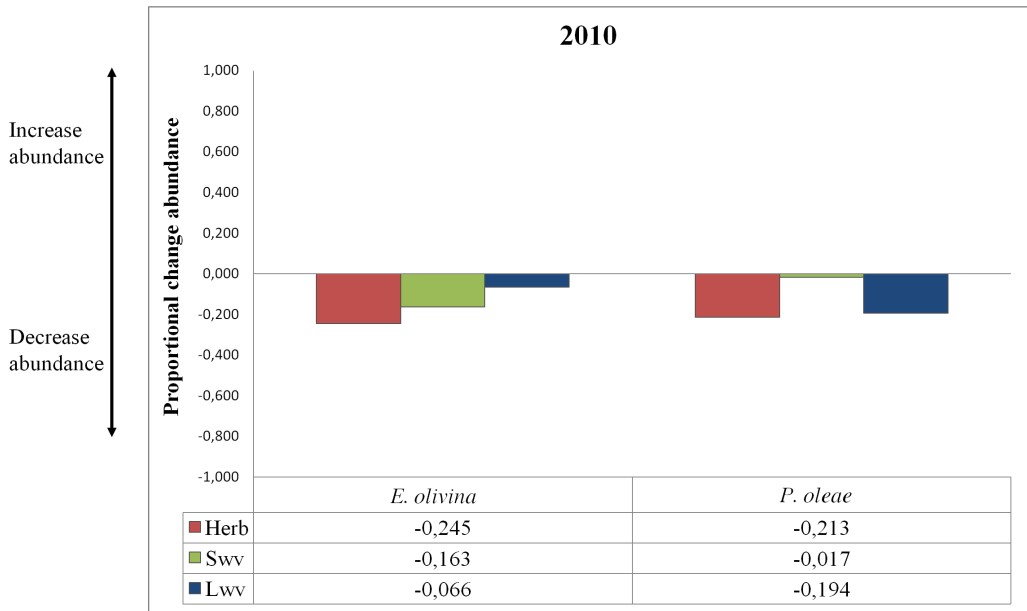

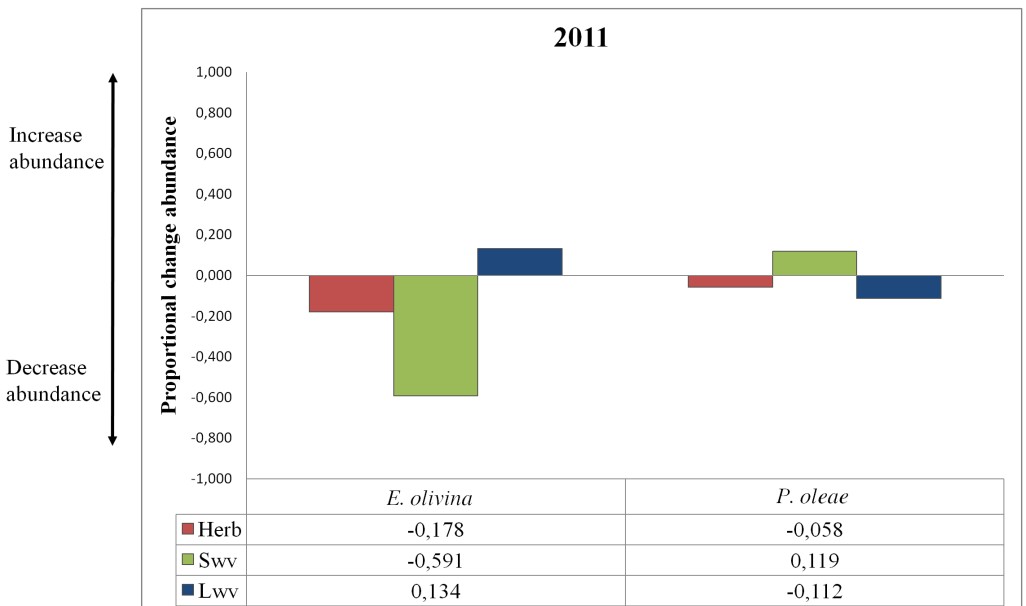

**Figure 2 Proportional change of abundance of the pests under the influence of different types of non-crop vegetation for both years of the study.**

appears later than *E. olivina* this could explain why small woody vegetation affects more consistently to *E. olivina* and large woody vegetation to *P. oleae*.

Reductions in pest numbers are likely to be associated with enhanced densities of predatory Heteroptera in olive plots as a response to herbaceous and small woody vegetation (*Paredes, Cayuela & Campos, 2013*) as these pests, especially *E. olivina,* are highly specific for olive and changes to their numbers would have nothing to do with emigration. Members of this taxon, such as *Anthocoris nemoralis* or *Deraeocoris punctum,*

have been described as predators of both *E. olivina* and *P. oleae* (*Kidd et al., 1999*; *Morris et al., 1999b*; *Scutareanu et al., 1999*; *Agustí, Unruh & Welter, 2003*). The variability observed between years in the proportional change of abundance of the two pest species might suggest an important role of climatic conditions in modulating pest responses to their environment. Changes in temperature and humidity could alter the phenology of pests and natural enemies and, therefore, influence insect population growth rate (*Wolda, 1988*; *Logan, Wolesensky & Joern, 2006*), which might ultimately change the effectiveness of natural enemies in controlling pest abundance from one year to the next.

There was no effect of inter-row ground covers and this is consistent with the results reported by *Aldebis et al. (2004)* and *Rodriguez, Gonzalez & Campos (2009)*. Notably, inter-row ground covers were found to increase the abundance of different groups of natural enemies including spiders, parasitoids and ants in an earlier study of this system (*Paredes, Cayuela & Campos, 2013*) but the present results indicate a lack of top down effects on *P. oleae* and *E. olivina*. This could reflect a lack of synchrony between these herbivore species and the natural enemies that exhibited a response to ground cover vegetation. Studies in other systems have indicated the significance of synchrony for pest suppression (*Fagan et al., 2002*; *Perdikis, Fantinou & Lykouressis, 2011*). This means an increase in the abundance of natural enemies promoted by ground cover occurs at a time of the year when pest abundance is low. This could be the case for spiders, which were enhanced by the ground covers (*Paredes, Cayuela & Campos, 2013*) but reach their maximum abundance by mid-August. This contrasts with the nymphs of *E. olivina* and the eggs and larvae of *P. oleae* that reach their maximum densities much earlier, in April or May. Thus spiders are responding to prey availability rather than suppressing population increase. Alternatively, the species of natural enemies favoured by ground cover might not be those that utilise *E. olivina* and *P. oleae* as prey. Parasitoids clearly fall within this category, since most of those reported to be enhanced by ground cover (*Paredes, Cayuela & Campos, 2013*) are not specific for the study pest species. Parasitoids of *E. olivina* and *P. oleae* mainly belong to families Encyrtidae and Elamisdae which, together, represent less than 30% of the abundance of those reported from this study system. In fact, 50% of the parasitoids belonged to the hymenopteran family Scelionidae, which mostly attack the eggs of spiders (*Fitton, Shaw & Austin, 1987*) and chrysopids in olive groves (*Campos, 1986*). This Intra-guild predation would weaken top-down forces in olive groves. Related to this effect, the increase in abundance of ants reported by *Paredes, Cayuela & Campos (2013)* might have a detrimental effect on other natural enemies in olive groves (*Morris et al., 1999a*; *Pereira et al., 2004*). In summary, even though ground cover enhanced the populations of some groups of natural enemies, this did not lead to reduce pest abundance so is not an optimal form of vegetation for use in olive conservation biological control.

This study has indicated the relative suitability of various types of non-crop vegetation in conservation biological control of *E. olivina* and *P. oleae* and provides an example of how such studies can help growers achieve the sometimes difficult task of balancing the practicalities of conserving appropriate forms of vegetation in ways that cause little or no disruption to normal agronomic practices (*Gurr et al., 2005*). *Prays oleae* is currently

the principal insect pest in olive groves in the Mediterranean Europe so it needs to be the primary target of management. Against this pest, herbaceous vegetation and large woody vegetation adjacent to olive crops provided the most consistent level of suppression. Small woody vegetation within olive groves appears less suitable as it gave modest reductions or increases in *P. oleae* densities. In areas where *E. olivina* is more likely to be the primary pest, small woody vegetation within olive groves offers more value, especially since its presence led to major reductions in the study year during which it was more common. Inter-row ground covers are shown to be unsuitable because, though these promote various natural enemy taxa, they do not provide suppression of either of the major pests. A longer-term study would be important to validate these tentative recommendations and identify precisely the underlying ecological process that can influence the success of olive pest conservation biological control.

## ACKNOWLEDGEMENTS

We thank Belen Cotes, Mario Porcel, Rafael Alcalá, Herminia Barroso and Luisa Fernández for their field and laboratory assistance; Juan Castro for his management of the experimental olive grove.

### Funding

This study was funded by Junta de Andalucía (PO7-AGR-2747) and the Ministry of Education of Spain (D Paredes FPU grant AP-2007-03970). The funders had no role in study design, data collection and analysis, decision to publish, or preparation of the manuscript.

### Grant Disclosures

The following grant information was disclosed by the authors:
Junta de Andalucía: PO7-AGR-2747.
Ministry of Education of Spain: FPU grant AP-2007-03970.

### Competing Interests

Dr. Geoff Gurr is an Academic Editor for PeerJ.

### Author Contributions

- Daniel Paredes conceived and designed the experiments, performed the experiments, analyzed the data, contributed reagents/materials/analysis tools, wrote the paper.
- Luis Cayuela analyzed the data, contributed reagents/materials/analysis tools, wrote the paper.
- Geoff M. Gurr wrote the paper.
- Mercedes Campos conceived and designed the experiments, wrote the paper.

## Supplemental Information

Supplemental information for this article can be found online at http://dx.doi.org/ 10.7717/peerj.116.

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
