# Peer review of "Effect of non-crop vegetation types on conservation biological control of pests in olive groves"

_PeerJ, doi:10.7717/peerj.116_

## Round 0.1 · original submission · Major Revisions

The comments from the reviewers are thoughtful, and consideration of these suggestions will lead to an improved manuscript. My primary concern is that there is an article referenced in press (Paredes D, Cayuela L, Campos M. 2013. Synergistics effects of ground cover and adjacent
natural vegetation on the main natural enemy groups of olive insect pests.
Agriculture, Ecosystems & Environment. In press.) that may have overlap with the present manuscript. I would appreciate a copy of this manuscript prior to a final decision on the present manuscript.

Reviewer 1 ·

Basic reporting

see below

Experimental design

see below

Validity of the findings

see below

Additional comments

Peer.Review
V2013:05:495 15 May 2013
Title: Effect of non-crop vegetation types on conservation biological control of pests in olive groves

General Comments
The paper presents an interesting approach to characterizing the impact of various vegetation types on two key pests of olive tree in the Western Basin of the Mediterranean. The study is a first step, exploratory effort at understanding how vegetation can affect abundance of two key pests. The study should be published but I have a number of problems with how the results are presented. Specific comments follow.

1. The field study does not incorporate hypothesis testing, and circumvents an experimental design that allows for hypothesis testing; there are no controls. This is understandable considering the scope of the vegetation that is under study: adjoining woody vegetation, interspersed woody vegetation, and then herbaceous ground covers. Using correlative techniques, and model fitting (all of which I’m not familiar), the authors appear to do a good job in showing how densities of olive moth and olive psylla are associated with different vegetation types. However the technique seems somewhat flawed in that the modeling is basically dependent on correlations between population size (both peak, mean, and variance) and time. Or perhaps these techniques somehow control for autocorrelation? One would expect there to be high R-values since the populations are growing through time (autocorrelation). Further, I don’t believe its correct to write that a particular vegetation ‘causes’ either of the pest populations (or natural enemies) to be higher or lower. Results are purely based on degree of association. No cause and effect was measured.

2. Results from this study would have been far more meaningful if authors had done some kind of cage studies, or malaise trapping in conjunction with the modeling work. Or provided more information on the taxa of natural enemies, their dynamics. And included climatic effects. This would be the most likely next step? One of the key natural enemies of the olive psyllid is a highly specific parasitoid, Psyllaephagus euphyllurae. Authors allude to this but never provided any data on its role, presence, impact. There is also a hyperparasitoid of this primary parasite. Most likely more of a generalist, and dependent on vegetation type, impact by Alloxysta sp. would vary based on the mixing and proximity of different vegetation types. Then there are climatic effects, which certainly have an impact on olive psyllid, independent of vegetation type. The psyllid is far more common the closer one gets to the coast line.

3. I thought olive fruit fly was considered one of, if not the worst pest of olives throughout the Mediterranean Basin. Yet the authors barely mention its role in this paper. Seems they should at least mention why they chose not to focus on this pest. Perhaps because the tephritids are such strong fliers or already under some kind of regional control program?


















Specific Comments
line comment
81 800 million Euro economic loss for the European Union, Spain, world. Which?
74 I believe Euphyllura olivine is restricted to the western Mediterranean basin. Does not include Middle East.
109 ‘former treatment’ refers to which one?
111 Add ‘by’ in front of ‘150 m’
112 Change ‘give’ to ‘provide’
159,161 Autocorrelation between insect abundance and time
194 Change ‘worse’ to ‘worst’
206-208 This analysis can only work for the ground cover , yes? Since no controls of this type were provided for the other vegetation types.
228 Since olive psyllid is highly specific for olive (can’t reproduce on anything else), changes to its numbers would have nothing to do with emigration. All to do with what’s eating it. This should be brought into the discussion.
240, and throughout Vegetation is not causing changes to abundance of target pests. Must be an indirect effect. So much more realistic to say various types of vegetation are associated with changes in pest numbers. Vegetation is affecting the numbers of various kinds of insects. Probably generalists, perhaps other competitors, which in turn affect moth and psyllid. Then there’s weather, climate.
282 Included in this discussion should be hyperparasitism of Psyllaephagus
286 Change ‘reduce’ to ‘reduced’
293 Change ‘pests’ to ‘pest’
300 Change’ thought’ to ‘though’
Fig. 1 caption Add ‘hypothetical distribution’
Fig. 2. Can’t make out bare vs. cover in image. Perhaps color will make difference
Fig. 3. caption Mention what ‘herb’, ‘Swv’ and ‘Lwv’ refer to in caption




G

·

Basic reporting

No Comments

Experimental design

No Comments

Validity of the findings

No Comments

Additional comments

This is a well written easy to follow manuscript that I thoroughly enjoyed reading.
I would have enjoyed seeing some information about any natural enemies collected in the field.
In the discussion, I would like to see a clarification in relation to what the reduced pest abundance was seen for example on line 240.
Apart from that this is a great manuscript showing research that attempts to move conservation biological control into a new cropping system.

Some minor changes would be:
Line 33: add ‘of pests’ after natural enemies
Line 60: add ‘in’ after …diversity for use ‘in’ olive grove…
Line 101: change to same time as that of the E. olivine nymphs
Line 126-127: change ‘This community was mainly compounded by…’ to This community consisted mainly of...’
Line 293 add 'it' between so and needs

Reviewer 3 ·

Basic reporting

See below

Experimental design

See below

Validity of the findings

See below

Additional comments

Conservation biological control (CBC), which has been recently great concerned, is considered to be an effective way to manage insect pest populations in an environmentally friendly manner. It usually involves attempts to provide resources to benefit natural enemies by environmental modification, which is usually associated with plant diversification. Plant diversification could, in many cases but not all, result in increased abundance of natural enemies and/or decreased abundance of insect pests. Scientists came to realize that identification of the ‘right’ diversity was more important than the diversity in per se. To learn the characteristics of local non-crop vegetations in CBC is the first step to identify the ‘right’ functional plants. Because local non-crop vegetations can adapt to local climate, and be easily planted as well as easily get access to, cheaper than endemic plants, they have the great potential to be used in CBC. Therefore, this paper is favored by people, especially olive producers, who are interested in CBC through rational management of farm plants.

This study provided a good example showing that increasing the abundance of natural enemies did not necessarily lead to the reduction of insect pests density, though data on natural enemies were documented in another paper. Hence, we must be cautious to cases that only take natural enemies into account in CBC.

In addition, this paper was well organized and written, and the data analysis seemed to be novelty and interesting.

Some specific comments/suggestions:

(1) Lines 30-31: “the recent ruling by the European Union to suspend the use of neonicotinoid insecticides”. I suggest that the references should be cited to support this statement.

(2) Lines 60-61: “use” should be replaced by “use in”.

(3) Line 101: “that” should be replaced by “as”.

(4) Line 109: It seems to me that “twelve” should be written as “12”.

(5) Line 243: “E. olivina” should be replaced by “E. olivina”.

(6) Lines 271-273: It is better to state the specific stage of the two insect pests when their abundances reach the peak in April or May. So “This contrasts with the E. olivina and P. oleae that…” could be replaced by “This contrasts with the nymphs of E. olivina and the eggs and larvae of P. oleae that…”

(7) Line 275: “prey” should be replaced by “as preys”.

(8) Line 415: The reference entitled “Synergistics effects of ground cover and adjacent natural vegetation on the main natural enemy groups of olive insect pests” has been published, the citation should be updated.

(9) I don’t think figure 1 is necessary, readers could easily understand the periods that insect pests appear by reading text only.

(10) There are four types of non-crop vegetation patches to be compared by sampling 12 plots. Generally, we would like to utilize randomized complete block design (RCBD) with four treatments and three blocks. Why was RCBD not applied? Additionally, why was olive grove separated into two parts, one with bare soil and another with ground cover?

(11) The authors used aerial photographs to measure the distance between natural vegetation patches and sampling plots. The distance measured in this way, however, reflects the horizontal distance between the two points only, which is shorter than the distance for which arthropods have to move when the two points were in different altitude. Figure 2 showed that the olive grove is located in mountain areas. Can this measure affect the results?

(12) Why did the authors choose 120 m as a standard deviation for the function? As you stated that this is the maximum dispersal distance of natural enemies but not the two insect pests mentioned in this paper.

(13) About the equation in line 187. What do the parameters of d.herb, d.lwp, d.swp stand for and how did you get the value of them? How did you get the value of Xherb, Xlwp and Xswp, which refer to density while you just measure the distance? Why can Xherb (density) by d.herb (reflects the effect of herbaceous density on pest abundance) represent the effect of natural vegetation patches on the maximum abundance of natural enemy groups?

(14) Does the proportional change in pest abundance calculated in this paper imply that there is a significant difference between certain type of vegetation patch and bare soil?

(15) What are the possible reasons for the large woody vegetation to consistently reduce the abundance of P. oleae in the two years? This should be discussed in the text.

(16) Readers can not distinguish the color of the columns in Figure 3, though they could judge from the value in the table.

(17) Tables 1 and 2 should be modified to meet the form of three-line table. In Table 2, year should be written in the same column and the authors should tell readers what do the words “EoN” and “PoFg” mean?

(18) Though model selection has been used in ecology, it is not familiar for a few readers. So I suggest the authors show relevant codes running in R programme in the supporting information or appendix.

---

## Round 0.2 · accepted · Accept

Thank your for the thorough and timely response to reviewer's comments.